# Discrepancy in Therapeutic and Prophylactic Antibiotic Prescribing in General Dentists and Maxillofacial Specialists in Australia

**DOI:** 10.3390/antibiotics9080492

**Published:** 2020-08-07

**Authors:** Cheryl Chen, Nicole Gilpin, Laurence Walsh

**Affiliations:** Faculty of Health and Behavioural Sciences, The University of Queensland School of Dentistry, UQ Oral Health Centre, 288 Herston Road, Herston, QLD 4006, Australia; n.v.gilpin@gmail.com (N.G.); l.walsh@uq.edu.au (L.W.)

**Keywords:** antibiotic prophylaxis, prescription behavior, dentists, dental specialists, oral and maxillofacial surgeons, oral surgery, dental extraction, dental implants

## Abstract

There are concerns that general dentists (GDs) and dental specialists may be prescribing antibiotics inappropriately. This study explored the prescribing habits and decision-making processes of GDs versus oral and maxillofacial surgeons (OMFSs). A case-based online questionnaire was used to examine the prescribing of therapeutic and prophylactic antibiotics in two clinical scenarios. Stratified and systematic sampling strategies were implemented to provide a representative sample. The final valid sample was 60 GDs and 18 OMFSs. The majority of OMFSs (61.1%) routinely prescribed antibiotics for the surgical removal of third molars, which was significantly greater than for GDs (23.5%). For implant placement procedures, 72.2% of OMFSs and 62.1% of GDs prescribed antibiotics. Amoxicillin was the most selected agent for both scenarios. All OMFSs would prescribe antibiotic prophylaxis for patients with uncontrolled diabetes mellitus in both cases, but only 56.0–63.0% of GDs would do this. GDs based prescribing decisions primarily on information from prescribing guides, while OMFSs relied more on information gained from specialist training. Surgical prophylaxis protocols differed considerably between groups. Both groups used surgical prophylaxis for some situations that are outside current recommendations. Education with regards to discrepancies between clinical practice and current guidelines for antimicrobial therapy is needed to progress antimicrobial stewardship.

## 1. Introduction

Antibiotic resistance has become a serious global public health concern due to the increasing prevalence of infections resulting from multidrug-resistant bacteria [1,2,3]. When analyzing the causal factors leading to the increasing incidence of antimicrobial-resistant bacteria, it has been established that the irrational prescribing and misuse of antibiotic medications, especially broad-spectrum agents, are some of the main drivers behind the emergence of resistant derivatives [1,2,3,4]. These species of bacteria can become resistant to the full armamentarium of antibiotics presently available, and are associated with rising morbidity and mortality rates from infections, accompanied by greater health costs at the individual and hospital levels [5,6]. Consequently, these dire outcomes associated with the use of antibiotics have encouraged researchers to investigate the antibiotic prescribing practices of dentists.

Antibiotics are the most commonly prescribed antimicrobial agents in dentistry [7]. Globally, dentists have been reported to prescribe up to 11.3% of all antibiotics [5,7]. Although, in general, antibiotic prescribing rates are typically lower in dentistry than those in medical practice [5], in recent years, reports of overprescribing within the dental community have emerged, both in Australia and overseas, despite growing concerns over antibiotic misuse [2,5,8,9,10].

Dental practitioners prescribe antibiotics for both therapeutic and prophylactic reasons. However, indications for antibiotic treatment in dentistry are limited, as the majority of dental infections can be treated effectively with direct operative measures [2,4,7,11]. Reliance on antibiotic therapy alone, particularly when using broad-spectrum rather than specific antibiotic agents, may not only result in a failure to control an infection, but it also risks the development of resistant bacterial strains and associated adverse effects for the individual patient [1,2]. Alternatively, the use of mesenchymal stem cells or dental pulp stem cells has been purported by some authors as a novel approach for use in surgical applications to enhance wound healing processes and avoid immunological reactions with xenogenic material through immunomodulatory and anti-inflammatory effects [12,13].

Prophylactic use of antibiotics in dentistry is intended to prevent metastatic infections (e.g., bacterial endocarditis) or the local or systemic spread of infection [7,14]. In modern dentistry, placement of dental implants to replace missing teeth has become a common procedure, with a long history of antibiotics being used as surgical prophylaxis. This practice, however, was developed before the era of antibiotic resistance recognition, with a recent systematic review by Park et al. [15] finding antibiotic use in healthy patients for the prophylaxis of surgical infection to be superfluous, as the practice does not appear to improve clinical outcomes based on the results of available double-blinded randomized controlled trials.

Correspondingly, it has been suggested that dentists may be prescribing antibiotics in situations where they are not necessary [1]. Although there are numerous studies reporting on the antibiotic prescribing patterns of medical specialists, there has been little research on the prescribing habits of dentists [16,17]. Little is known regarding disease-specific dental antibiotic use [17]. Internationally, within several countries, there have been studies into the prescribing patterns of dentists. These report that there is a tendency to prescribe broad-spectrum antibiotics empirically, either to provide prophylaxis or to manage oral/dental infection [1,8,16,18,19,20,21,22,23]. Several studies documenting the routine use of antibiotics during dental implant placement found that most dentists used a rule-based approach, rather than considering the need for antibiotics on a case-by-case basis [24,25,26,27,28]. This is at odds with the systematic review findings of Park et al. [15] that stressed the principles of antimicrobial stewardship and avoiding the use of antibiotics as a routine measure in healthy patients.

When looking at Australian dentists’ antibiotic prescribing practices specifically, Jaunay et al. [8] conducted a postal survey in 2000 of approximately 10% of South Australian general dental practitioners. Respondents lacked knowledge in some key areas, such as the incidence of adverse reactions to antibiotics, the risk of development of multi-resistant strains, and the indications for the appropriate use of prophylactic antibiotics, paralleling the findings of studies overseas on dental prescribing practices [8]. More recently, an Australian study conducted by Ford et al. [2] showed that dentally prescribed antibiotics dispensed to concessional beneficiaries had increased by some 50% over a 12-year period from 2001 to 2012. A similar study published by Teoh et al. [10] reported a 15.3% increase in the prescription of the broad-spectrum combination product amoxicillin/clavulanic acid from 2013 to 2016, suggesting that Australian dental practitioners may not be prescribing these medicines appropriately.

While the aforementioned studies report on the prescribing patterns and trends of dentists as a prescribing group, the research on clinicians’ knowledge, attitudes, and behaviors regarding antibiotic use is limited. Results from previous studies assessing medical physicians indicate that patient demands and expectation, providers’ perceptions related to patient expectations, fear of litigation, and diagnostic uncertainty are important factors that may lead to overprescribing [29,30]. Similar pressures may be affecting prescribing practices in the dental field. Therefore, it is important to investigate the prescribing habits and decision-making of dental practitioners, particularly in the context of recorded increases in the prescription of antimicrobial medicines by dentists over recent years [2,8].

The present study was designed to compare the prescribing habits of general and specialist dental clinicians practicing within Australia, and to explore practitioners’ rationale for antibiotic prescribing, in addition to the factors behind their prescribing practices for select dental procedures and patient circumstances.

## 2. Results

A total of 243 general dentists (GDs) and 68 oral and maxillofacial surgeons (OMFSs) were invited formally to participate in the survey. The final valid sample was 60 GDs and 18 OMFSs, which represented 0.3% of GDs and 8.1% of OMFSs in all of Australia.

### 2.1. Demographics

GD practitioners surveyed graduated from their primary dental degree between 1964 and 2015. OMFS practitioners obtained their primary dental degrees between 1959 and 1998. The majority of GDs (91.7%) and OMFSs (82.4%) obtained their primary dental degree in Australia. A wide range (5 to 48 years) of specialist practice was recorded for OMFS, with an average of 23 years.

Workplace environments differed significantly between GDs and OMFSs. GDs predominantly worked in the private sector only (55 GDs, 91.7%), while OMFSs were split between a public-private sector combination (11 OMFSs, 64.7%) and private sector only (6 OMFSs, 35.3%). There were no OMFSs who worked in the public sector only (0.0%) while 3 GDs worked in the public sector only (3.9%) and 2 worked in a public-private combination (2.6%).

Participant characteristics are recorded in Table 1.

### 2.2. Antibiotic Prophylaxis and Antimicrobial Regimens for Surgical Procedures in Non-Medically Compromised Patients

#### 2.2.1. Prescribing Frequency

Table 2 shows the antibiotic prophylaxis and antimicrobial regimens routinely employed by surveyed practitioners for specific clinical procedures in non-medically compromised patients with no reported allergies. Significant differences were observed in the frequency of routine antibiotic prescription between the GD and OMFS groups for the surgical extraction of third molars (*p* < 0.01). The majority of OMFSs (61.1%) routinely prescribed antibiotics for the surgical removal of third molars; this frequency was significantly lower for GDs (23.5%). In implant placement procedures, the frequency of prescription was high for both the GD (62.1%) and OMFS (72.2%) groups.

#### 2.2.2. Preferred Timing of Antibiotic Administration

If selected, the administration of antibiotics was predominantly chosen to be postoperative only. This was the case for both the surgical procedures of third molar removal (60.9%) and implant placement (38.7%). There were no significant differences between the GD and OMFS groups for preferred timing of administration. The timing of administration alternatives chosen were preoperative only, perioperative only, and combinations of preoperative or perioperative antibiotics in addition to postoperative administration.

The preferred timing of antibiotic administration for practitioners who prescribed antibiotics are recorded in Table 3.

#### 2.2.3. Preferred Antibiotic Agent

Amoxicillin used alone was the preferred antibiotic agent for the majority of practitioners who chose to prescribe antibiotic agent/s in both case scenarios. The alternative antibiotic agents chosen were amoxicillin/clavulanic acid, phenoxymethylpenicillin, cephalexin, and metronidazole plus amoxicillin in combination.

The preferred antibiotic agent/s for practitioners who prescribed antibiotics are recorded in Table 4.

#### 2.2.4. Justification

The most frequently selected justifications for the prescription of antibiotics for third molar removal were ‘better healing’ (43.5%) and ‘routine protocol’ (39.1%). Likewise, ‘routine protocol’ (50.0%) was the most common rationale for the use of antibiotic prophylaxis in implant placement. Contrarily, the reasoning for practitioners who did not normally prescribe was often that antibiotics were of ‘no benefit/not indicated’ for the procedure.

#### 2.2.5. Antiseptic Disinfection

There were no significant differences between practitioner groups in the frequency of antiseptic disinfection prior to treatment for the investigated procedures.

### 2.3. Prophylactic Antibiotics for Medically Compromised Patients

Table 5 shows the medical conditions for which practitioners might consider prescribing prophylactic antibiotics in relation to the selected procedures investigated in the clinical case scenarios.

Significant (*p* < 0.05), very significant (*p* < 0.01), and extremely significant (*p* < 0.001) statistical differences were observed between the GD and OMFS practitioner groups’ antibiotic prophylactic frequencies for medically compromised patients.

For third molar surgical extraction, all OMFSs (100.0%) opted to provide antibiotic prophylaxis for patients with uncontrolled diabetes mellitus. This difference was extremely significant when compared to the GD group, where just over half (56.0%) would prescribe antibiotic prophylaxis (*p* < 0.001). OMFSs were also more likely to prescribe antibiotic prophylaxis to patients who were HIV positive (*p* < 0.05) or had a history of injecting drug use (*p* < 0.01).

In implant dentistry, all OMFSs (100.0%) would again prescribe antibiotic prophylaxis for patients with uncontrolled diabetes mellitus. Similarly, this was a significantly higher frequency than for the GD group (63.0%) (*p* < 0.01). OMFSs were also more likely to prescribe antibiotic prophylaxis for patients with a history of intravenous bisphosphonate therapy (*p* < 0.05).

### 2.4. Factors Affecting Practitioners’ Prescribing Practices

Preferred resources for GDs and OMFSs differed substantially. Figure 1 shows the reported most important factor or resource affecting surveyed practitioners’ prescribing decisions, displayed by the percentage of each practitioner group. Almost half of GDs (47.4%) used a prescribing guide as their first-line resource to facilitate their decision to prescribe antibiotics and the choice of agent in clinical situations, while no OMFSs (0.0%) ranked prescribing guides as their number one resource. The majority of OMFSs (53.3%) preferred to rely on information gained from their specialist training.

Of those who used a prescribing guide in their practice (60/60 GDs, 16/18 OMFSs), the *Monthly Index of Medical Specialties* (*MIMS*) was the most popular in the OMFS group, with 68.8% of OMFSs selecting the guide. This guide was also highly selected for GDs (60.0%). The most widely used prescribing guide for GDs was the *Australian Therapeutic Guidelines* (*ATG*) *Oral and Dental*, with 80.0% of GD respondents using the guide in their clinical practice. This was significantly higher than the OMFSs’ use of this guide (43.8%) (*p* < 0.01). The more recently published *ATG Antibiotic*, however, was used by fewer GD practitioners (25.0%), with a two-fold uptake in the OMFS group (50.0%). Similarly, the *Australian Medicines Handbook* was not a commonly selected guide for the GD group (8.3%), but uptake was significantly higher (31.3%) with OMFSs (*p* < 0.05). Data on the types of prescribing guides used by practitioners are displayed in Figure 2.

### 2.5. Adverse Drug Reactions

Over half of the OMFSs (52.9%) had previously observed adverse drug reactions to antibiotics that required medical referral. This was significantly higher compared to the GD group, where only 10.0% of practitioners had observed these reactions. Reactions observed by respondents included anaphylactic reactions to penicillin/amoxicillin and cephalosporin, tetracycline allergies, skin rashes, erythema multiforme, and death.

## 3. Discussion

This study investigated the use of therapeutic and prophylactic antibiotics by general dental practitioners and oral and maxillofacial surgeons with respect to third molar surgical extractions and dental implant placement. Discrepancies were observed between the antibiotic protocols of different practitioner groups for both case scenarios. The majority of OMFSs (61.1%) opted to prescribe antibiotics for surgical third molar extraction. This was significantly greater than the GD group (23.5%) (*p* < 0.01). For single dental implant placement, most OMFSs (72.2%) and GDs (62.1%) prescribed antibiotics routinely. The differences between the antibiotic protocols of the GD and OMFS practitioner groups may be attributed to different training and varied resources. Many practitioners who prescribed antibiotics cited ‘better healing’, a reduction of post-operative infection, or ‘routine protocol’, while those who did not prescribe claimed ‘no benefit/not indicated’. These two schools of thought are in stark contrast to one another, and this may reflect in part incongruences in the current literature and in guidelines that are used globally.

Prophylactic antibiotic therapy is defined as “the administration of any antimicrobial agent that prevents the development of disease” [31], and its use in oral surgical procedures remains contentious. As surgical procedures in the mouth are considered to be “clean-contaminated surgery” [32], there is potential for infection of the surgical site. In 2004, Rajasuo et al. [33] reported that surgical extractions of partly erupted mandibular third molars caused bacteremia of a higher frequency and lasting a longer duration than assumed by previous studies. The study found that 88% of patients had detectable bacteremia, with 31 species cultured, consisting predominantly (74%) of anaerobes. For 44% of the patients, bacteremia occurred immediately after extraction, and for 50%, within one minute after the initial incision [33]. The percentage of patients with bacteremia tended to drop as the time of surgery progressed [33].

Perioperative prophylactic antibiotic usage is consistent with the established guidelines used for clean-contaminated procedures in general surgery, and can be justified [32]. However, the incidence of complication rates may be too low to warrant routine prophylactic antibiotic usage for the removal of third molar teeth. Poeschl et al. [34] cited a complication incidence of 1–5.8%, with deep space infections only arising in cases with preoperative pericoronitis. Other authors estimated the infection rate of lower third molars to be even lower at less than 1% [31,35]. Furthermore, postoperative administration of antibiotics, which was favored by study respondents, may have limited benefit in terms of the outcomes of third molar surgery [34].

While prophylactic antibiotic administration may have the potential to reduce the impact of detectable bacteremia, it must be remembered that antibiotic agents are not without possible adverse effects. The benefits of antibiotic usage must outweigh the risks. In the present study, a variety of serious adverse reactions were reported to have been observed by participants. Systemic complications with antibiotic usage include anaphylaxis, development of resistant strains, interaction with other medicines, and side effects, including nausea, vomiting, diarrhea, and pseudomembranous colitis [22,31]. These adverse effects are a major cause of morbidity and mortality for patients [11,36]. The frequent comment from practitioners of ‘better healing’ associated with the use of antibiotics noted in this survey may reflect a tendency to prescribe antibiotics, despite risks of adverse effects, in an attempt to aid wound healing post-surgery. Considering this phenomenon, recent advances into the novel use of mesenchymal stem cells in vitro show promise for their application in clinical and surgical environments. Regenerative medicine may be a possibility for use as a tissue healing aid in the future, perhaps without the drug-related consequences of antibiotic use [12,13].

A compromised immune system is considered to be the most important factor in predicting the likelihood of postoperative infection [31]. The medical status of a patient, including their age, nutrition, diabetes, obesity, and other infections, has been found to impact wound infection rates. Additionally, surgical factors may influence the likelihood of infections, including prolonged operating time, a contaminated or dirty field, the failure to eliminate dead space, and extensive tissue trauma [31,36]. Therefore, although significant bacteremia may be induced during third molar surgery, prophylactic antibiotics may not be required for the majority of patients, as bacterial invasion is unlikely to overwhelm host defenses in the healthy patient, and establishment of infection is rare [15].

Similarly, in terms of dental implant placement, a recent systematic review by Park et al. [15] in 2018 based on double-blinded randomized controlled trials available in the current literature concluded there was no statistically significant benefit of the use of prophylactic antibiotics for the prevention of post-operative infections, and should be avoided in healthy patients. While some retrospective studies show a beneficial effect of perioperative antibiotics on tissue healing and reduction of the levels of bacteria at the site of insertion, these studies have several limitations, including a retrospective design as well as the lack of a placebo group [15].

The subjective nature of treatment outcomes and the definition of ‘infection’ means it is difficult to compare outcomes between studies investigating the benefits of antibiotic prophylaxis of oral surgical procedures. Clinical symptoms of pain, swelling, and trismus may be unreliable. Few studies use objective markers, such as the serum levels of acute-phase proteins, compounding this problem [37]. Further prospective objective trials, with relevant control groups, are needed to resolve this situation.

A large percentage of practitioners routinely used antiseptics for implant placement in the present study, with no significant disparity between the GD and OMFS groups. Two-thirds of OMFSs also used antiseptic rinses prior to surgical third molar removal. The choice of antiseptic was not investigated in this study. A study by Barbosa et al. [38] in 2015 found that a 0.2% chlorhexidine mouthwash before tooth extraction significantly reduced the post-extraction bacteremia as compared to the control at 15 min post-extraction. The authors suggested that the practice may be extended to other procedures in dentistry. It is possible that antiseptic protocols have the potential to effectively shift dentoalveolar ‘clean-contaminated’ surgery closer to the realm of ‘clean’ surgery. A reduction in the levels of bacteria in the oral cavity and/or gingival crevice may be more effective in preventing infection than the use of high-dose systemic antibiotics in reducing an already established bacteremia. Prevention is certainly optimal, and antiseptic solutions are generally well-tolerated with markedly fewer adverse effects than systemic antibiotics [38]. Such a change in practice would, however, need to be supported by randomized controlled trials.

The global epidemic of coronavirus disease 2019 (COVID-19) has also highlighted the use of antimicrobials in dentistry, with preoperative antiseptic rinses recommended to reduce the viral load of the etiologic agent, severe acute respiratory syndrome coronavirus 2 (SARS-CoV-2), in the oral environment [39]. However, the viral load contained in human saliva is very high and may not be eliminated with mouthwashes, so further personal protection measures and avoiding or minimizing operations producing droplets or aerosols should be followed to protect the health and safety of practitioners, staff, and patients [39].

In the present study, the largest differences between the prescribing practices of respondent groups were in the use of antibiotics for medically compromised patients. While overall the results figures are consistent with findings from previous studies, including Jaunay et al. [8] and Palmer et al. [20], some disparities between practitioner groups were observed in this study. In general, for surgical dental procedures, OMFSs were much more likely to prescribe antibiotic cover to immunocompromised patients (e.g., patients with uncontrolled diabetes mellitus) for protection against bacterial infection. Key cardiac conditions of prosthetic heart valves were identified strongly; however, immunocompromising conditions of previous head and neck radiotherapy and poorly controlled diabetes mellitus were less readily identified by GDs as medical conditions requiring protective cover against surgical site infections. This outcome was similar to findings from a study on South Australian dentists in 2000 [8].

One important area of contention is surgical prophylaxis for patients with orthopedic implants. The majority of the GD (59.3%) and OMFS (76.5%) groups opted to prescribe antibiotics for these patients. Late infections of joint replacements are primarily associated with skin commensal staphylococci and β-hemolytic streptococci; these microorganisms are rarely isolated from dentally related bacteremia, as they do not form part of the normal oral flora [20,40]. The relatively high numbers of GDs and OMFSs wanting to prescribe here may reflect ignorance of the current guidelines (which advise against prescribing in this situation) or advice from overly cautious orthopedic surgeons.

When considering the agent of choice, the present study found amoxicillin to be the overwhelming favorite of most respondents for all scenarios. These findings parallel those of both local and foreign studies of dental prescribing [5,8,18,19,23,41,42]. It is also consistent with the studies by Ford et al. [2] and Teoh et al. [10] of the prescribing patterns of Australian dental practitioners, where amoxicillin was found to be the most commonly prescribed antibiotic, accounting for 66% of all prescriptions by dentists in 2012 and 64.3% in 2016. Interestingly, the preference of amoxicillin by Australian dentists is not reflective of the guidance provided by the *ATG* [43] for odontogenic infections. The *ATG* [43] recommendation is for phenoxymethylpenicillin, a narrow-spectrum penicillin effective against 85% of oral bacteria (mainly Gram-positive organisms), as the drug of choice. Amoxicillin is a semi-synthetic antibiotic derived from the penicillin nucleus, and has a somewhat broader spectrum, including a Gram-negative spectrum similar to ampicillin [2,8]. There is increasing evidence that the broader spectrum of amoxicillin may be contributing to the development of resistant strains in non-oral pathogens [8]. In addition, amoxicillin is more commonly associated with adverse effects, such as gastrointestinal problems and rashes [2]. There is little clinical difference in the action of these two agents, aside from dosing regime, and the microbiological advantage is moot. It has been suggested that the increase in use of amoxicillin may be attributed in part to the pharmaceutical industry as it has been marketed as a superior product over penicillin [8,22]. Despite this, it remains the clinician’s responsibility to prescribe the most suitable agent for each clinical indication, whilst taking into account the risks of adverse reactions and the development of resistant and multi-resistant strains.

Practitioners’ preferred reference to support and shape their antibiotic prescribing habits was explored in the present study. The differences between the prescribing practices of the GD and OMFS practitioner groups may be attributed to differences in training, the preferred use of different prescribing guides and resources, and the varied rank of importance given to the various prescribing resources. The results of this study show a variety of responses for practitioners’ opinions of the most important factor behind their prescribing practices. General dental practitioners were most likely to refer to a prescribing guide as their first-line reference for antibiotic prescribing, while OMFSs were more likely to refer to their specialist training. This may explain why OMFSs were more likely to routinely prescribe antibiotics for surgical prophylaxis than other groups. OMFSs may be more considerate or cognizant of general surgical principles for clean-contaminated surgery [32], in line with their specialist surgical training. Incongruences in the literature may have downstream effects on training, resources, and guidelines both locally and internationally, and may be resulting in practitioner differences when prescribing antibiotics. Following the completion of this study, the *Australian Therapeutic Guidelines* (*ATG*) *Oral and Dental* was updated in late 2019 [44]. This updated prescribing guide may relieve some confusion where disparities between the previous issues of *ATG Oral and Dental* [43] and *ATG Antibiotic* [45] might have led to divergent clinical practices. Further research should be conducted into the clinical use and effectiveness of antibiotics, ensuring that there are relevant control groups. Practitioner education should be based on high-level evidence, such as randomized controlled trials, prospective studies with objective markers, and meta-analyses of homogenous studies.

Defined as the appropriate, safe, and discriminate use of antimicrobials for therapy and prophylaxis, antibiotic stewardship has been championed by governments, international health organizations, and initiatives to reduce issues associated with antibiotic-resistant pathogens [4,11]. With regards to following the principles of antibiotic stewardship, dental practitioners may be falling short. A reported 66% of antibiotics prescribed in a dental setting are not clinically indicated [11]. In the present study, one-third of practitioners prescribed antibiotics routinely for third molar surgical extraction. This percentage rose to almost two-thirds for dental implant placement. For practitioners who routinely prescribed antibiotics, there was a preference for amoxicillin, a moderate-spectrum antibiotic agent, which may not be in line with antimicrobial stewardship principles. Dosage regimens, considered to impact on antibiotic stewardship [11], were not investigated in the present study.

The results of this study provide a macroscopic broad overview of the prescribing practices and rationale of registered practicing dentists in Australia. It is, to date, the largest exploratory study into the prescribing habits of dental practitioners to be completed in Australia. The inclusion of different practitioner groups, with a large proportion of specialists, allows significant comparisons to be drawn between GDs and specialists. Many other studies internationally, with even larger sample sizes, lacked involvement of or comparison with specialists [5,18,20,22,23,24,25]. Nevertheless, the analysis of current clinicians’ prescribing habits over a number of different procedures and medical conditions must be viewed in the light of certain limitations. Antibiotic prescription is largely recommended as a case-by-case activity, and this study was not able to inspect these clinical nuances, including procedural complexity or patient compliance. These variables may be addressed in futures studies by considering a more defined research focus. Additionally, the final small number of responding subjects in the OMFS group raises questions as to how representative their responses may be of their cohort.

The combination of explorative clinician-based research to understand the status quo and controlled investigations into the impact of antimicrobials on procedural outcomes may inform subsequent practitioner education. Further studies are needed to explore how well dental practitioners are aware of drug interactions and other consequences of excessive prescribing. The streamlining of training, professional development, resources, and guidelines is essential for clinicians to deliver minimized risk and evidence-based dentistry to patients in the era of antibiotic resistance.

## 4. Materials and Methods

A cross-sectional survey was conducted on registered general dentists (GDs) and dental specialists using an online case-based questionnaire incorporating clinical case scenarios for prophylactic and therapeutic prescription.

For comparing GDs and OMFSs, two clinical case scenarios formed the basis of the questionnaire: One for third molar surgical extraction and another for single dental implant placement. Both procedures were completed under local anesthetic only (case scenarios are reproduced in Appendix A.)

For each scenario, respondents were surveyed on their antibiotic prescribing practices for these procedures and patient variables. They were asked to indicate whether they would prescribe antibiotics, their agent/s of choice, timing of antibiotic administration, and rationale behind their decision in each proposed situation. The conditions were selected based on published Australian guidelines [43,45]. Participants were permitted to skip scenarios that did not pertain to their current clinical practice. Additional data collected include general knowledge about antibiotics, reporting of adverse drug reactions, use of recently published antibiotic guidelines, and demographic data, including the type of practice, country and year of graduation from primary dental degree, and specialty training (if applicable).

The questionnaire was run as a pilot on a small group of general and specialist dental practitioners (who were not part of the final sample) to test the clinical case scenarios for applicability and ease of understanding, as well as to evaluate the readability and reduce any ambiguity of wording before distribution. The final survey operated via the SurveyMonkey^®^ website for participant access and data collection.

The dental practitioner directory published by the Australian Dental Association (ADA), *The dental directory: 2016 edition* [46], was applied as the sampling frame for the study population of GD and OMFS practitioners. Practitioners listed in the directory have opted in to have their practice contact details included. Clinicians identified as general dental practitioners (GDs) and oral and maxillofacial surgeons (OMFSs) were contacted to participate in the study. Only currently practicing clinicians were contacted. Stratified and systematic random sampling strategies were implemented to reduce bias and provide a representative sample across Australia. Using the directory [46], practitioners were pooled into GD and OMFS categories and ordered by postcode of practice across the states of New South Wales, Victoria, and Queensland, beginning at 2000, 3000, and 4000 respectively. GDs were sampled systematically using a sampling interval of three. All eligible OMFSs were contacted. Potential participants were contacted by the practice details listed in the directory [46]. Clinicians whose practices/practitioners agreed to receive further participation information were formally invited to the survey via an email with a link to the SurveyMonkey^®^ online survey and the participant information sheet attached. A total of 243 GDs and 68 OMFSs were invited. Two follow-up emails were sent to all respondents. Informed consent was obtained for all participants at the commencement of the online survey and prior to their inclusion in the study.

Participants accessed the survey through a provided link to the online survey, hosted on the SurveyMonkey^®^ website. Data from the survey was collected by the SurveyMonkey^®^ web form and downloaded for analysis after the close of the survey. The survey link was open for a period of four months. All survey responses were collected and stored in a de-identified form.

Data was coded and reported using Microsoft^®^ Excel software. Descriptive statistics were computed and comparative analysis between groups performed using Chi-squared and Fisher’s exact tests. A value of *p* < 0.05 was considered statistically significant.

This study was approved by The University of Queensland School of Dentistry Ethics Committee, and conducted in full accordance with the regulations governing experimentation on humans (Project reference number: 1813).

## 5. Conclusions

Variations in the prescribing habits between general dentists and specialists for routine procedures within the fields of oral surgery and implant dentistry were observed, especially with regards to antibiotic prophylaxis for medically compromised patients. Surgical prophylaxis protocols differed between clinician groups. Practitioners were found to rank resources guiding the prescription of antibiotic agents differently, which may have an impact on their clinical practice. Practitioner education with regards to discrepancies between clinical practice and current guidelines for antimicrobial therapy should be addressed. Ultimately, there remains a crucial need for clear evidence-based guidelines on the prescribing of antibiotics by dental practitioners, especially for surgical prophylaxis, to reduce the inappropriate use of systemic antibiotics in dentistry.

## Figures and Tables

**Figure 1 antibiotics-09-00492-f001:**
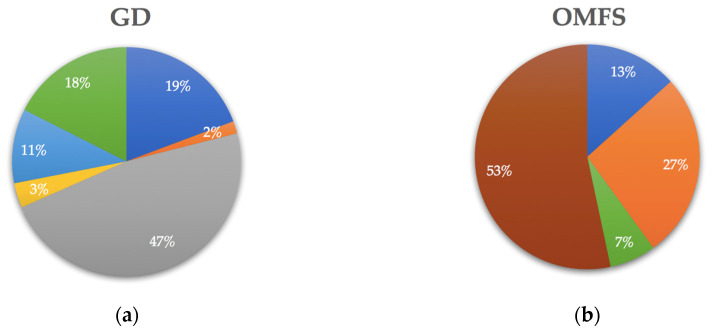
Most important factor or resource affecting practitioners’ prescribing decisions. (**a**) GD group (*n* = 60); (**b**) OMFS group (*n* = 18); (**c**) Key.

**Figure 2 antibiotics-09-00492-f002:**
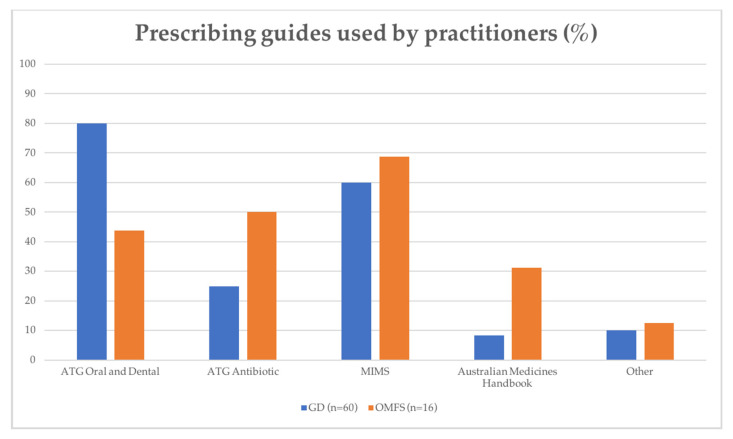
Percentage (%) use of prescribing guides within GD (*n* = 60) and OMFS (*n* = 16) groups.

**Table 1 antibiotics-09-00492-t001:** Participant characteristics.

Characteristics	All	GDs	OMFSs	Significance (*p* Value) ^3^
	(*n* = 77)	(*n* = 60)	(*n* = 17) ^1^	
**Primary dental degree**				
Graduation (year) ^2^				-
Mean	1990.1	1992.6	1981.6	
SD	13.1	12.4	12.5	
Range	1659–2015	1964–2014	1959–1998	
Obtained in Australia, *n* (%)				*p* > 0.05
Yes	69 (89.6)	55 (91.7)	14 (82.4)	
No	8 (10.4)	5 (8.3)	3 (17.7)	
**Specialist experience**				
Years of practice experience as a specialist (years)	-	-		-
Mean			23	
SD			12.5	
Range			5–23	
**Workplace environment**				
Private sector only	71 (79.2)	55 (91.7)	6 (35.3)	*p* < 0.001
Public sector only	3 (3.9)	3 (5.0)	0 (0.0)	
Both private and public	2 (2.6)	2 (3.3)	11 (64.7)	

^1^ One OMFS respondent opted not to disclose demographic details. ^2^ One GD respondent did not disclose their year of graduation from their primary dental degree. ^3^ Calculated using the Chi-squared test.

**Table 2 antibiotics-09-00492-t002:** Surgical procedures for which practitioners prescribe therapeutic antibiotics and the antibiotics agents used for patients with no relevant medical history and no allergy to penicillin.

Procedure	Respondent Type		Disinfection of Oral Cavity (%)	Prescribed Antibiotics (%)	Most Frequent Timing of Antibiotic Administration	Most Frequently Prescribed Antibiotic Agent
**Third molar surgical extraction**	Total	*n* = 69	49.3	33.3	Postoperative only 60.9%	Amoxicillin only 78.3%
	GDs	*n* = 51	43.1	23.5 ^1^		
	OMFSs	*n* = 18	66.7	61.1 ^1^		
**Single implant placement**	Total	*n* = 47	87.2	66.0	Postoperative only 38.7%	Amoxicillin only 77.4%
	GDs	*n* = 29	93.1	62.1		
	OMFSs	*n* = 18	77.8	72.2		

^1^ Very significant finding: Greater percentage of OMFSs would prescribe antibiotics than GDs (*p* < 0.01). Calculated using Fisher’s exact test.

**Table 3 antibiotics-09-00492-t003:** Preferred timing of antibiotic administration of practitioners who prescribed antibiotics.

Procedure	Respondent Type		Preferred Timing of Antibiotic Administration (%)				
			Preoperative Only	Perioperative Only	Postoperative Only	Pre- and Postoperative	Peri- and Postoperative
**Third molar surgical extraction**	Total	*n* = 23	21.7	4.2	60.9	4.3	8.7
	GDs	*n* = 12	9.1	9.1	63.6	9.1	9.1
	OMFSs	*n* = 11	33.3	0.0	58.3	0.0	8.3
**Single implant placement**	Total	*n* = 31	25.8	9.7	38.7	12.9	12.9
	GDs	*n* = 18	15.4	15.4	46.2	7.7	15.4
	OMFSs	*n* = 13	33.3	5.6	33.3	16.7	11.1

**Table 4 antibiotics-09-00492-t004:** Preferred antibiotic agent/s of practitioners who prescribed antibiotics ^1^.

Procedure	Respondent Type		Preferred Antibiotic Agent/s (%)				
			Amoxicillin Only	Phenoxymethylpenicillin Only	Cephalexin Only	Metronidazole Plus Amoxicillin	Amoxicillin/Clavulanic Acid
**Third molar surgical extraction**	Total	*n* = 23	78.3	8.7	4.3	4.3	0.0
	GDs ^1^	*n* = 12	75	8.3	0.0	8.3	0.0
	OMFSs	*n* = 11	81.8	9.1	9.2	0.0	0.0
**Single implant placement**	Total	*n* = 31	77.4	0.0	6.5	9.7	6.5
	GDs	*n* = 18	83.3	0.0	5.6	11.1	0.0
	OMFSs	*n* = 13	69.2	7.7	7.7	7.7	15.4

^1^ One GD did not specify their preferred antibiotic agent in the third molar surgical extraction case scenario.

**Table 5 antibiotics-09-00492-t005:** Medical conditions for which practitioners provide antibiotic prophylaxis.

Medical History	Third Molar Surgical Extraction		Single Implant Placement	
**% of practitioners providing antibiotic prophylaxis for the procedures listed**				
	GDs (*n* = 50)	OMFSs (*n* = 18)	GDs (*n* = 27)	OMFSs (*n* = 17)
Patients on oral anticoagulants or platelet inhibitors	4.0	5.6	11.1	17.7
HIV positive	24.0 ^1^	55.6 ^1^	37.0	47.1
History of injecting drug use	8.0 ^2^	38.9 ^2^	14.8	41.2
Chronic hepatitis B or C	14.0	33.3	22.2	41.2
Uncontrolled diabetes mellitus	56.0 ^3^	100.0 ^3^	63.0 ^2^	100.0 ^2^
Cardiac valve replacement or mitral valve prolapse	92.0	83.3	85.2	88.2
Prosthetic joint in past two years	48.0	72.2	59.3	76.5
History of head and neck cancer and radiotherapy	48.0	55.6	48.2	64.7
Oral bisphosphonate therapy for the past three years	66.0	83.3	63.0	82.4
History of intravenous bisphosphonate therapy	68.0	88.9	55.6 ^1^	88.2 ^1^

^1^ Significant finding (*p* < 0.05). Calculated using Fisher’s exact test. ^2^ Very significant finding (*p* < 0.01). Calculated using Fisher’s exact test. ^3^ Extremely significant finding (*p* < 0.001). Calculated using Fisher’s exact test.

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
