# Peer review of "Discrepancy in Therapeutic and Prophylactic Antibiotic Prescribing in General Dentists and Maxillofacial Specialists in Australia"

_antibiotics, 2020, doi:10.3390/antibiotics9080492_

Round 1

Reviewer 1 Report

The authors present result from a cross-sectional study regarding potential differences between general dentists and oral and maxillofacial surgeons. The use of antibiotics for dental procedures is highly important regarding antibiotic stewardship programs since they are often used for therapeutic and prophylactic reasons. The work is interesting and fitting for the journal and the subsection that is submitted. Some comments follow:

  1. I suggest a change in the title of the manuscript since there are additional specialties in dentist practice. Please indicate that the analysis is between general dentists and the OMFS. Also it would be optimum to state that the work refers to Australia since they may be differences in other countries and regions (Lines 69-78).
  2. Line 12 abstract: Please replace "various" with the actual number since two scenarios were used.
  3. 2.2.3. Apart of amoxicillin what other antibiotics were prescribed? A table in results or in appendix would be helpful. Also amoxicillin alone or amoxicillin/ clavulanic acid combination?
  4. Table 2. What was the timing of administration over the two scenarios? The regimens that are followed etc. (please edit table columns width)
  5. Table 3. What are the reasons of not prescribing? Are they stated in the survey?
  6. 2.4 Factors affecting practitioners’ prescribing practices (figure 1). Considering the evidence-based medicine hierarchy (EBM pyramid) in what level, each of this factors, is attributed with? This is crucial cause some conclusions could be extracted regarding the validity of the evidence behind these factors. Something is mentioned in lines 389 and 452 but I would recommend to add some comments in the results or discussion section.
  7. 2.5 What was the level of awareness regarding potential drug interactions? Are two groups question patients regarding co-administered drugs? If not in the survey I encourage authors to look further into it. Also if possible please include any percentages of the reported ADRs from the respondents. Any reason that can be extracted from the data behind this difference in ADRs report habits?
  8. Some additional discussion could be added over the results of this study. In authors opinion, what conclusion can be drawn from the analysis of the results regarding the compliance to the antibiotic stewardship protocols for DG and OMFs in Australia? Are they in bad/good/excellent level regarding principles of antimicrobial stewardship?
  9. Lines 377-378, needs mention of these studies to be comparable or rephrase.
  10. A minor comment for the online survey (mostly considering the tool that was applied in the study). How the participation of the dentist was ensured and not e.g. of an assistant, and whether the link may receive more than one answers (from different IPs). 

Reviewer 2 Report

The topic of this article entitled “Differences in Prescribing Habits of General Dentists versus Specialist Practitioners.” is aimed to assess the role of prescribing habits of general dentists compared to specialist pactictioners. It is an interesting topic and within the journal's scope. Nevertheless, this reviewer would suggest some improvements, before further considerations.  The study has certainly new information related to the role of therapeutic strategies and its control in oral and maxillofacial pathologies. Introduction: Authors have reported several important topics related to bacterial infections and resistance; however, poor has been reported on the role of resident stem cells, which can act as immunomodulatory and anti-inflammatory players in the local environment (Please, see and discuss: Ballini, A.; Scacco, S.; Coletti, D.; Pluchino, S.; Tatullo, M. Mesenchymal Stem Cells as Promoters, Enhancers, and Playmakers of the Translational Regenerative Medicine. Stem Cells Int 2017; 2017:3292810). In this light it’s important to briefly describe something about novel and “safe” in-vitro reparative/regenerative/therapeutic cell-based models, working without any additive (e.g. BSA) to apply safely such protocols in human models (Please, see and discuss “Highly Efficient In Vitro Reparative Behaviour of Dental Pulp Stem Cells Cultured with Standardised Platelet Lysate Supplementation. Stem Cells Int. 2016;2016:7230987.”)

Finally, authors should also give their point of view on COVID-19 in dental fields (please, see and discuss “COVID-19 Outbreak: An Overview on Dentistry. Spagnuolo G, De Vito D, Rengo S, Tatullo M. Int J Environ Res Public Health. 2020 Mar 22;17(6):2094. doi: 10.3390/ijerph17062094.”)

Reviewer 3 Report

Although, the topic addressed by authors is not novel, the antibiotic overuse is still an essential problem affecting the clinical dentistry and medicine all over the word, with a long-term direct impact on antibiotic resistance worldwide.

The presented and discussed results demonstrate a significant discrepancy between guidance and clinical practice, particularly in cases when dentists  often rely on their 'experience and intuition' despite of existing current recommendations for antibiotic prescribing. 

Minor comments:

1. Suggested title: Discrepancy in therapeutic and prophylactic antibiotic prescribing in general dentists and maxillofacial specialists.

2. Detailed description need re: selection of representative groups.

3. Final small number of subjects in MFOS group may not be really representative for statistic analysis. This has to be clearly underline as a weakness of study.

4. Rephrasing in discussion section, along with repetitions correction. 

I personally appreciate the efforts made by authors to highlight the global problem of unjustified prescriptions in dentistry.    

Round 2

Reviewer 2 Report

ALL THE COMMENTS HAVE BEEN ADDRESSED